# TRPM8′s Role in the Shift Between Opioid and Cannabinoid Pathways in Electroacupuncture for Inflammatory Pain in Mice

**DOI:** 10.3390/ijms252313000

**Published:** 2024-12-03

**Authors:** Dinh-Trong Pham, Rae-Mann Hsu, Mao-Feng Sun, Chien-Chen Huang, Yi-Hung Chen, Jaung-Geng Lin

**Affiliations:** 1Graduate Institute of Acupuncture Science, China Medical University, No. 91, Xueshi Road, North District, Taichung City 404328, Taiwan; phamdinhtrong1111@gmail.com; 2Faculty of Traditional Medicine, Hai Phong University of Medicine and Pharmacy, Hai Phong City 180000, Vietnam; 3School of Chinese Medicine, China Medical University, No. 91, Xueshi Road, North District, Taichung City 404328, Taiwan; 4International Master Program in Integrative Health, China Medical University, Taichung City 404328, Taiwan; 5Department of Chinese Medicine, An Nan Hospital, China Medical University, Tainan City 700, Taiwan; 6School of Post-Baccalaureate Chinese Medicine, College of Chinese Medicine, China Medical University, Taichung City 404328, Taiwan; 7Chinese Medicine Research Center, China Medical University, Taichung City 404328, Taiwan

**Keywords:** complete Freund’s adjuvant, inflammatory pain, mechanical allodynia, analgesia, electroacupuncture, TRPM8

## Abstract

The TRPM8 channel, a temperature-sensitive ion channel, plays a crucial role in various physiological processes, particularly in the modulation of inflammation and nociception. Although electroacupuncture (EA) is a recognized analgesic treatment for pain conditions, its interaction with TRPM8 remains underexplored. This study aims to determine TRPM8′s role in EA-induced analgesia using a murine model of inflammatory pain. Mechanical allodynia, evidenced by a reduced paw withdrawal threshold (PWT), was induced in both wild-type and *Trpm8^−/−^* mice through CFA injection. EA applied at the GB34 and LR3 acupoints significantly alleviated mechanical allodynia in both groups. In wild-type mice, the analgesic effects of EA were partially reversed by naloxone (an opioid receptor antagonist) or AM251 (a CB1 receptor antagonist) and fully reversed by their combination. In contrast, only AM251 reversed EA-induced analgesia in *Trpm8^−/−^* or TRPM8-inhibited wild-type mice (via AMTB treatment, a TRPM8 antagonist), indicating no involvement of the opioid pathway. Additionally, the combination of menthol, a partial TRPM8 agonist, and EA enhanced analgesia in wild-type mice. In *Trpm8^−/−^* or AMTB-pretreated mice, the CB1 receptor agonist WIN 55,212-2 (WIN) exhibited stronger analgesic effects compared to wild-type controls. These findings suggest that EA at LR3 and GB34 mediates analgesia through both opioid and endocannabinoid pathways. TRPM8 is critical for EA to activate the opioid pathway, while its inhibition or deletion shifts the analgesic mechanism towards reliance on the cannabinoid system. Understanding this mechanistic shift may help optimize EA treatment strategies and improve pain management outcomes.

## 1. Introduction

Transient Receptor Potential Melastatin 8 (TRPM8) is a thermosensitive ion channel with significant physiological and pathological implications, activated by non-noxious cooling temperatures (<23–28 °C) [1,2]. TRPM8 channels are distributed in a variety of human bodies: prostate tissue [3], the skin and mucosae [4], lung epithelial cells [5], and artery myocytes [6]. In the nervous system, TRPM8 is found in high levels within the trigeminal (TG) nerve, autonomic ganglia, and dorsal root ganglia (DRG) of the nervous system [7]. TRPM8 is highly expressed in sensory neurons, responding to harmful and harmless cold stimuli [8]. Furthermore, TRPM8 expression in sensory neurons increases considerably following nerve injury or inflammation [9]. TRPM8 has promising therapeutic applications in pain management and inflammatory disorders, as preclinical studies have demonstrated that its activation can produce analgesic effects [10,11]. The antinociceptive effect of menthol mediated through TRPM8 has been observed to depend on the natural opioid pathways of the body [12,13]. Specifically, the TRPM8-dependent analgesic effects of menthol are likely mediated by the activation of the central κ-opioid system, with minimal involvement of the µ and δ opioid systems [14].

In Chinese medicine, acupuncture is an ancient practice in which thin needles are inserted into specific body points to alleviate pain and treat diseases [15]. Acupuncture enhances endogenous opioid levels in the central and peripheral nervous systems, contributing to analgesia. Clinical and animal studies demonstrate that manual acupuncture/electroacupuncture primarily upregulates cerebrospinal fluid opioids [16,17]. Furthermore, studies have found that in both humans [18] and mice [19], the opioid receptor antagonist naloxone reduced the analgesic effects of acupuncture. When β-endorphin and enkephalin were purified in the early 1980s, these opioids were identified as important components of acupuncture in both humans and livestock [20,21]. These findings suggest that acupuncture treatment may enhance peripheral opioid activation.

Research examining the correlation between distinct levels of EA analgesia reveals that low-frequency (2 Hz) EA primarily induces the stimulation of met-enkephalin, β-endorphin, and endomorphin [20,21], with the level of β-endorphin significantly increased in the lumbar cerebrospinal fluid of patients with recurrent pain after EA treatment [20]. Conversely, high-frequency (100 Hz) EA stimulation specifically elicits a response from the opioid peptide dynorphin [22]. The analgesic effects of EA in inflammatory pain are linked to the CB1 system [23], and the antinociceptive impact of EA predominantly relies on CB1 receptors situated in the ventrolateral region of the periaqueductal gray [24]. Concurrently, the anti-inflammatory attributes of EA are associated with peripheral CB2 receptors [25]. The analgesic and anti-inflammatory benefits of EA in rats with temporomandibular joint (TMJ) arthritis are dependent on the activation of CB1 and CB2 receptors [26]. The results indicated that the antinociceptive of EA was reversed by AM251, a CB1 antagonist, and that AM630, a CB2 antagonist, reversed the anti-inflammatory of EA [26]. In addition, another study suggests that CB1 receptors play a role in EA-induced orofacial antinociception in rats exposed to noxious facial heat, confirming these findings [27]. In this study, AM251, but not AM630, blocked antinociception induced by EA alone at acupoint ST36 [27]. Pretreatment with an endocannabinoid metabolizing enzyme inhibitor (MAFP) or an anandamide reuptake inhibitor (VDM11), inhibitors of endocannabinoid metabolism and reuptake, respectively, prolonged and intensified EA’s antinociceptive effects, highlighting the involvement of CB receptors in EA’s mechanism of action [27].

The analgesic effects of acupuncture are mediated primarily through two mechanisms: the opioid and cannabinoid pathways. TRPM8 is a key receptor for analgesic pathways. However, the specific role of TRPM8 in acupuncture-induced analgesia remains to be fully understood. This study aims to elucidate the role of TRPM8 in the analgesic effects of EA, particularly in inflammatory pain.

## 2. Results

### 2.1. The Analgesic Effect of Electroacupuncture at GB34 and LR3 in CFA-Induced Inflammatory Pain

The mice in each group were acclimated to the observation area and tested using the von Frey test on the baseline day, CFA day, Day 1, and Day 2 (Figure 1B). On the baseline day, there was no significant difference in mechanical sensitivity, as measured by paw withdrawal threshold (PWT), between the five groups (Naive: 6.5 ± 0.35 g, CFA: 6.46 ± 0.37 g, CFA + EA: 6.1 ± 0.18 g, CFA + Sham-EA: 6.5 ± 0.24 g, CFA + Lido + EA: 6.4 ± 0.34 g). On the CFA day, the Naive group was injected with normal saline, and the CFA, CFA + EA, CFA + Sham-EA, and CFA + Lido + EA groups were intraplantarly injected with CFA (30 μg in 30 μL/mouse) after recovering from anesthesia in the observation box. The PWT substantially decreased after CFA injection (CFA: 1.28 ± 0.2 g, CFA + EA: 1.35 ± 0.2 g, CFA + Sham-EA: 1.27 ± 0.1 g, CFA + Lido + EA: 1.0 ± 0.22 g), while the Naive group, which was injected with saline, did not change (Naive: 6.5 ± 0.3 g). This indicates that CFA injection induced mechanical allodynia, which persisted throughout the experiment (Figure 1C).

On Days 1 and Day 2, the CFA + EA group received EA treatment at GB34 and LR3 for 20 min, 60 min before the von Frey test, while the Naive and CFA groups were under isoflurane anesthesia at the same time. During the 2-day EA treatment, the CFA + EA group on Day 1 (CFA + EA: 3.12 ± 0.22 g) and Day 2 (CFA + EA: 3.13 ± 0.12 g) had significantly increased PWT compared to the CFA group on Day 1 (CFA: 1.47 ± 0.21 g) and Day 2 (1.53 ± 0.11 g) (*p* < 0.05). However, the CFA + Sham-EA (Day 1: 1.27 ± 0.08 g, Day 2: 1.37 ± 0.14 g) group had no significant effect compared with the CFA group on the PWT (*p* > 0.99). These findings demonstrate that a 20-min low-frequency electrical stimulation at acupuncture points GB34 and LR3 produces a significant analgesic effect in CFA-induced inflammatory pain. Additionally, the CFA + Lido + EA group (Day 1: 1.65 ± 0.23 g, Day 2: 1.63 ± 0.29 g) did not exhibit an enhanced the PWT post-acupuncture treatment, showing no significant difference compared to the CFA group (*p* > 0.99). This suggests that the analgesic effect of EA at GB34 and LR3 is significantly inhibited by pretreatment with 2% lidocaine at GB34, administered 1 min before EA, as illustrated in Figure 1C and Appendix A.

### 2.2. Naloxone and AM251 Effectively Reversed the Analgesic Effect of EA in CFA-Induced Inflammatory Pain in Wild-Type Mice

In this experiment, mice were divided into four groups to investigate the mechanism of the analgesic effect of acupuncture on mechanical allodynia pain: Vehicle (saline) + EA group, Naloxone + EA group, AM251 + EA group, and Naloxone + AM251 + EA group (Figure 2A). On the Baseline and CFA days, the PWT was not significantly different among the four groups.

In the Vehicle + EA group, the PWT significantly increased after EA treatment for two consecutive days, suggesting the analgesic effects of EA. Both the cannabinoid antagonist (AM251) and opioid antagonist (naloxone) significantly reversed the analgesic effect of EA, as PWT decreased in the AM251 + EA group and Naloxone + EA group compared to the Vehicle + EA group on Day 1 (Naloxone + EA: 1.8 ± 0.3 g, AM251 + EA: 2.4 ± 0.2 g, Vehicle + EA: 2.95 ± 0.2 g) and Day 2 (Naloxone + EA: 1.92 ± 0.2 g, AM251 + EA: 2.4 ± 0.3 g, Vehicle + EA: 3.1 ± 0.2 g) (*p* < 0.05). Interestingly, the Naloxone + EA group had a more decreased PWT than the AM251 + EA group. This difference is significant when compared with the Naloxone group (*p* < 0.05) (Figure 2B and Appendix A).

Furthermore, the combination of naloxone, AM251, and EA (Naloxone + AM251 + EA group) further decreased the analgesic effect of EA, as evidenced by the further reduction in PWT. This was observed when compared with the Naloxone + EA group on Day 1 (Naloxone + AM251 + EA: 1.51 ± 0.1 g) and Day 2 (1.43 ± 0.12 g), with statistical significance (*p* < 0.05). In addition, the Naloxone + AM251 + EA group also significantly reversed the effect of EA in the PWT compared with the AM251 + EA group (*p* < 0.05). These results suggest that while EA reduced mechanical allodynia induced by CFA, this effect was reversed not only by the opioid receptor antagonist naloxone but also by the CB1 receptor antagonist AM251, as shown in Figure 2B. Additionally, the analgesic effect of EA was further reversed by the combined use of the cannabinoid antagonist AM251 and naloxone in CFA-induced inflammatory pain.

### 2.3. Effectiveness of TRPM8 Antagonist (AMTB) in Modulating EA Combined with AM251 or Naloxone Effects in CFA-Induced Inflammatory Pain in Wild-Type Mice

We used AMTB, a TRPM8 antagonist [28,29], to investigate the association of TRPM8 with the analgesic effect of EA. As shown in Figure 3A, AMTB was administered before EA treatment. On the baseline and CFA days, the PWT was not significantly different between the Vehicle (saline) + EA group and the AMTB + EA group. During two days of EA treatment, the effect of EA was maintained and PWT was increased in all groups (Vehicle + EA: 3.0 ± 0.2 g, AMTB + EA: 2.9 ± 0.3 g). However, there was no significant difference in PWT between the Vehicle + EA group and AMTB + EA group (Figure 3B and Appendix A), indicating that the TRPM8 antagonist (AMTB) did not affect the analgesic effect of EA in CFA-induced inflammatory pain in wild-type mice.

To further understand the involvement of TRPM8 in the mechanism of EA analgesia, mice were divided into three groups: AMTB + EA, AMTB + Naloxone + EA, and AMTB + AM251 + EA. As shown in Figure 3C and Appendix A, there was no significant difference in the PWT among the three groups on the baseline day. After CFA injection on the CFA day, PWT decreased in all groups. On Day 1 and Day 2 post-EA treatment, the effect of EA was reversed in the AMTB + AM251 + EA group (Day 1: 1.6 ± 0.2 g, Day 2: 1.5 ± 0.4 g) compared with the AMTB + EA group (Day 1: 2.9 ± 0.1 g, Day 2: 3.0 ± 0.3 g) (*p* < 0.05). However, there was no significant difference in PWT between the AMTB + EA group and the AMTB + Naloxone + EA group (Day 1: 3.2 ± 0.4 g, Day 2: 3.1 ± 0.3 g). These results suggest that when mice were pretreated with AMTB, the analgesic effect of EA was reversed by the CB1 receptor antagonist but not by the opioid receptor antagonist.

### 2.4. The Effect of EA on the Pretreatment of CB1 Receptor Antagonist (AM251) or the Opioid Receptor Antagonist (Naloxone) in CFA-Induced Inflammatory Trpm8^−/−^ Mice

We further used *Trpm8^−/−^* mice to evaluate the association of the analgesic effect of EA and TRPM8. The *Trpm8^−/−^* mice were tested with the von Frey test on baseline day, CFA day, Day 1, and Day 2 (Figure 4). On the baseline PWT results, there was no significant difference in the PWT between the *Trpm8^−/−^* mice and wild-type mice. After CFA injection on the CFA day, the *Trpm8^−/−^* mice showed a dramatic reduction in PWT (1.6 ± 0.2 g) in all experiments, as shown in Figure 4. During Days 1 and 2, when EA treatment was applied, the CFA + EA group exhibited a significant increase in PWT (Figure 4A, Appendix A) (Day 1: 3.6 ± 0.2 g, Day 2: 3.8 ± 0.2 g) compared with the CFA group (Day 1: 1.6 ± 0.3 g, Day 2: 1.7 ± 0.1 g) (*p <* 0.05). This suggests that EA also reduced inflammatory pain in the *Trpm8^−/−^* mice.

We then used *Trpm8^−/−^* mice to investigate how TRPM8 affects the mechanism of EA analgesia. In Figure 4B,C and Appendix A, we included two additional groups: the Naloxone + EA group and the AM251 + EA group. Figure 4B demonstrates that the PWT in the Naloxone + EA group over two consecutive days was not significantly different compared to the Vehicle + EA group. This indicates that naloxone did not reverse the analgesic effect of EA in the mechanical allodynia induced by CFA in *Trpm8^−^^/^^−^* mice. In contrast, Figure 4C shows that the effect of EA was reversed in the AM251 + EA group, as the PWT was reduced compared with the Vehicle + EA group (*p <* 0.05). This indicates that the CB1 receptor antagonist (AM251), but not naloxone, can reverse the effect of EA in mechanical allodynia induced by CFA in *Trpm8^−^^/^^−^* mice.

### 2.5. Menthol Increased the Analgesic Effect of EA in CFA-Induced Inflammatory Pain in Wild-Type Mice but Not Trpm8^−/−^ Mice

Menthol typically targets TRPM8, a specific type of channel that allows calcium to pass through and is expressed in a portion of sensory neurons located within the dorsal root ganglion and trigeminal ganglion [14]. In 2006, the application of menthol and icilin to activate the TRPM8 receptor, whether applied topically or intrathecally in animal models, was shown to reduce pain [9]. Recently, menthol’s potent analgesic properties have demonstrated effectiveness against formalin-induced flinching and inflammatory hypersensitivity, albeit at higher dosages [30]. This study aims to evaluate the analgesic effect of the combination of EA and menthol.

Menthol was administered orally (20 mg/kg) before EA treatment in both wild-type mice and *Trpm8^−/−^* mice (Figure 5 and Appendix A). As shown in Figure 5C, in wild-type mice, the Menthol + EA group exhibited a greater increase in PWT compared to the Vehicle + EA group (*p <* 0.05). This suggests that menthol enhances the analgesic effect of EA in mechanical allodynia induced by CFA.

In the *Trpm8^−/−^* mice (Figure 5B), after menthol pretreatment on Day 1 and Day 2, the effect of EA still increased the PWT in the Menthol + EA group, but there was no significant difference compared with the Vehicle + EA group. This result indicates that EA reduced the mechanical allodynia induced by CFA in the *Trpm8^−/−^* mice, and menthol did not affect the analgesic effect of EA in these mice.

### 2.6. Cannabinoid Receptor Type 1 (CB1 Receptor) Agonist, WIN 55,212-2, Induced the Analgesic Effect in Inflammatory Pain in Both Wild-Type Mice and Trpm8^−/−^ Mice

In this experiment, we tested WIN 55,212-2 [31,32] as a CB1 receptor agonist for relieving inflammatory pain (Figure 6 and Appendix A). The timeline is shown in Figure 6A. We first elucidated the effect of WIN 55,212-2 in mechanical allodynia and the influence of AMTB and TRPM8 knockout on its effect.

Wild-type (WT) mice were divided into three groups: Vehicle + WIN, AMTB + WIN, and AM251 + WIN. Figure 6B shows that administration of WIN 55,212-2 decreased pain by increasing the PWT in the Vehicle + WIN group over two days of treatment. However, pretreatment with AMTB in the AMTB + WIN group further enhanced the effect of WIN compared to the Vehicle + WIN group in terms of PWT (*p <* 0.05). Furthermore, when AM251 was used as a pretreatment in the AM251 + WIN group, the effect of WIN was significantly attenuated in terms of PWT compared to the Vehicle + WIN group (*p <* 0.05). This indicates that pretreatment with AMTB could enhance the analgesic effect of WIN 55,212-2 in mechanical allodynia induced by CFA.

In Figure 6C, WIN 55,212-2 was tested in both WT mice and *Trpm8^−/−^* mice. Mice were divided into three groups: Vehicle + WIN in wild type, Vehicle + WIN, and AM251 + WIN in *Trpm8^−/−^* mice. On the CFA day, all groups exhibited a decrease in PWT. On Day 1 and Day 2 of WIN treatment for pain, PWT increased in the WT Vehicle + WIN group. However, in the *Trpm8^−/−^* AM251 + WIN group, PWT was significantly decreased compared to the WT Vehicle + WIN group (*p <* 0.05).

We also compared the effects of WIN 55,212-2 in both WT mice and *Trpm8^−/−^* mice to test whether the effect is CB1 receptor-dependent. The mice were divided into three groups: Vehicle + WIN in wild-type mice, and Vehicle + WIN and AM251 + WIN in *Trpm8^−/−^* mice. On the CFA day, all groups showed a decrease in PWT. On Day 1 and Day 2, when WIN was used to treat the pain, there was an increase in PWT in the WT Vehicle + WIN group. However, in the *Trpm8^−/−^* Vehicle + WIN group, PWT increased more than in the WT Vehicle + WIN group (*p <* 0.05). Additionally, pretreatment with AM251 in *Trpm8^−/−^* mice greatly decreased PWT in the *Trpm8^−/−^* AM251 + WIN group (*p <* 0.05).

These data indicate that the analgesic effect of WIN 55,212-2 in *Trpm8^−/−^* mice was greater than in wild-type mice and that the effect of WIN 55,212-2 is CB1 receptor-dependent.

## 3. Discussion

The TRPM8 receptor, a thermoreceptor distributed throughout the human body, including the nervous system [7] and skin [4], is crucial in modulating pain, particularly in inflammatory conditions [10,11]. EA has been well-documented in experimental studies for its efficacy in pain relief [25,33]. This study aimed to explore the interaction between EA and TRPM8 in the context of inflammation-induced pain. The initial findings revealed that the analgesic effect of EA was reversed by pre-treatment with naloxone or AM251. When both agents were used in combination, the analgesic effect of EA was completely abolished, suggesting that EA at acupoints GB34 and LR3 exerts its effects primarily through the opioid receptor and its associated pathway, with additional involvement of the cannabinoid system. However, in *Trpm8^−/−^* mice or in wild-type mice where TRPM8 was inhibited via AMTB treatment, the EA-induced analgesic effect was only reversed by AM251, highlighting the requirement of TRPM8 in activating the opioid pathway by EA. When TRPM8 is inhibited or deleted, the analgesic mechanism of EA shifts toward reliance on the cannabinoid system. Additionally, in a CFA-induced inflammatory pain model, the PWT was significantly higher in *Trpm8^−/−^* mice or in the presence of a TRPM8 receptor inhibitor (AMTB) when WIN was administered, suggesting that the analgesic effect of WIN is enhanced in the absence of TRPM8 under inflammatory conditions. Finally, the combination of menthol and EA produced a greater increase in PWT compared to either treatment alone, indicating that activation of TRPM8 potentiates the analgesic effects of EA in inflammatory pain. These findings provide novel insights into the mechanisms of EA, emphasizing the pivotal role of TRPM8 in modulating pain pathways and offering a foundation for developing more effective treatments. This study is the first to report these mechanistic insights, contributing valuable knowledge to the field of global pain management.

### 3.1. Constitute a Pain Model and Evaluate the Mechanism of EA Treatment in the Mechanical Allodynia Induced by CFA

CFA is a drug commonly used to induce inflammation in inflammatory pain research [34,35,36]. CFA injection into the plantar hind paws of mice triggered the release of pain mediators in inflammatory tissue, mimicking human myofascial pain [37,38]. The von Frey test was employed to measure PWT levels after CFA injection. The PWT decreased at 60 min after injection of CFA, consistent with previous studies [35,36,38]. Besides using mechanical methods to assess pain levels in mice post-CFA injection, other tests such as the hot plate test and radial heat test are also employed in some studies [12,38]. These results are consistent with previous research and validate our pain measurement methodology.

As is well known, the GB34 acupoint is frequently used to treat numerous diseases. GB34 has been reported to be useful in Parkinson’s disease treatment and pain relief. In Parkinson’s disease, GB34 enhances motor functions, increases the levels of dopaminergic fibers and neurons in the striatum and the substantia nigra [39], improves the survival of dopaminergic neurons, and inhibits α-syn expression linked to dopaminergic cell loss in the substantia nigra [40]. Additionally, GB34 reversed CCI-induced mechanical allodynia and thermal hyperalgesia and decreased Glucose transporter-3 (GLUT-3) protein expression in the left medial prefrontal cortex (mPFC) [41]. Combining EA with milnacipran (5 μL) at GB34 results in a stronger antiallodynic and antihyperalgesic effect [42]. EA at GB34 and GB30 significantly attenuates thermal hyperalgesia and mechanical allodynia, as well as upregulates precursor proopiomelanocortin (POMC) mRNA and endorphin protein levels in inflamed skin tissues [43]. In addition, the LR3 acupoint is commonly used in clinical practice for treating various diseases, including brain and inflammatory diseases [44,45]. Stimulation at LR3 can activate or deactivate brain areas associated with vision, movement, sensation, emotion, and pain relief [46].

In the present study, the inflammation location on the plantar surface of the hind paw was also very close to the LR3 point, suggesting that LR3 was chosen as well as a local point to enhance the effect of acupuncture for pain relief. GB34 acts as the distal acupoint, while LR3 serves as the local acupoint to enhance the impact of EA. Additionally, neurosurgery has shown that the deep fibular nerve traverses both GB34 and LR3 [47,48]. This anatomical relationship also facilitates a more straightforward interpretation of the neurological mechanisms underlying the analgesic effects of these acupoints. For these reasons, GB34 and LR3 were selected as the acupoints for this study.

EA’s analgesic effect in peripheral inflammation-induced hyperalgesia was mediated not only by the opioid receptor pathway [49] but also by the cannabinoid system pathway [25]. The analgesic effect of EA on paw pressure threshold was prevented in carrageenan-induced inflammatory pain rat models by intraplantar injection of naloxone or selective antagonists against μ, δ, and κ opioid receptors before EA treatment [50,51]. Furthermore, intraplantar naloxone methiodide in the CFA-induced thermal hyperalgesia paradigm eliminated the alleviation of EA pain as measured by paw withdrawal latency [52]. Researchers have found that acupuncture needles placed at the PC6 acupoint (EA-PC6) produce analgesia by activating CB1 and OX1 receptors [24]. Additionally, AM251 injected into the ventrolateral periaqueductal gray matter (vlPAG) of mice with knee osteoarthritis reduced chronic knee osteoarthritis pain by upregulating CB1 receptors and 2-AG levels; the effect of EA on pain hypersensitivity was reversed by AM251 [53]. The efficacy of EA has diminished mechanical and thermal sensitivity. Furthermore, elevated expression and levels of CB1 in the DRG indicate that EA works via the peripheral endocannabinoid system [54]. However, the CB1 receptor antagonist AM251, administered into the acupoint prior to acupuncture, abolished the effect of EA [54].

Previous acupuncture research has identified the cannabinoid pathway as a significant underlying mechanism [23,25]. In the current study, the opioid receptor antagonist naloxone and CB1 receptor antagonist AM251 suppressed the analgesic effect of EA sessions in CFA-induced inflammatory pain in wild-type mice. In particlular, EA’s analgesic effect in *Trpm8^−/−^* mice or under TRPM8 inhibition in WT mice was reversed only by the CB1 receptor antagonist AM251, not by the opioid receptor antagonist naloxone. These data indicate that the beneficial effect of EA at GB34 and LR3 involved the underlying pathways in mechanical hyperalgesia mediated via the opioid receptor pathway and the CB1 receptor pathway activation. However, EA is not able to activate the opioid receptor pathway if TRPM8 is suppressed or if TRPM8 knockout is performed. It has been suggested that TRPM8 receptors play an important role in the analgesia pathway of EA via the opioid system.

### 3.2. Menthol May Enhance the Analgesic Effect of EA via TRPM8 Receptor Activation

TRPM8 receptor, primarily known for sensing cold and activated by compounds like menthol, has emerged as a significant modulator of pain and inflammation, with complex interactions across opioid and endocannabinoid pathways. A subset of sensory neurons in the trigeminal ganglion and dorsal root ganglion is sensitive to menthol’s effects because it blocks calcium from entering a specific type of channel called TRPM8 [14]. Activation of TRPM8 leads to calcium influx, which is critical for neurotransmitter release and synaptic transmission, thereby impacting pain signaling [55]. It was confirmed that chronic morphine enhances cold hypersensitivity by increasing excitability and reducing desensitization in TRPM8-expressing neurons by activating the μ-opioid receptor-protein kinase C beta (MOR-PKCβ) signaling pathway [56]. In a previous study, activation of μ-opioid receptors leads to internalization of TRPM8, while knockout of TRPM8 leads to a reduction in morphine-induced cold analgesia [57]. The above reports indicate the synergistic effects between TRPM8 and opioid receptors.

Additionally, TRPM8 shows intricate crosstalk with the endocannabinoid system, where endogenous cannabinoids like anandamide (AEA) and N-arachidonoyl dopamine (NADA) act as functional agonists, reducing TRPM8 activity and modulating cold and pain sensations [58]. Phytocannabinoids such as cannabidiol (CBD) and Δ9-tetrahydrocannabinol (THC) also influence TRPM8; CBD, in particular, acts as a potent agonist, potentially desensitizing TRPM8 and modifying sensory responses [59]. The above reports indicate the antagonizing effects between the cannabinoid system and TRPM8, which are consistent with the present study.

In our study, EA combined with menthol significantly reduced mechanical pain more than EA alone in wild-type mice but not in *Trpm8^−/−^* mice. Using naloxone, a nonselective opioid receptor antagonist [13,57], and the specific κ-opioid antagonist, nor-NBI, significantly reduced the TRPM8-dependent analgesic effects on both acute and inflammatory pain [14]. In contrast, the selective μ-opioid receptor antagonist CTOP, as well as the δ1-opioid receptor antagonist 7-benzylidenenaltrexone (BNTX) and the δ2-opioid receptor antagonist naltriben, failed to inhibit menthol-induced antinociception. These results indicate that TRPM8-mediated analgesia induced by menthol is primarily dependent on activation of the central κ-opioid receptor system, with little to no involvement of the μ- or δ-opioid receptor systems [14].

Our findings show that menthol, as an agonist of TRPM8, can enhance the analgesic effects of EA via the opioid receptor pathway. This evidence suggests that TRPM8 is involved in the opioid receptor-mediated analgesia pathway of EA.

### 3.3. The CB1 Receptor Agonist (WIN 55,212-2) Reduced the Mechanical Allodynia Induced by CFA in the Wild-Type Mice and Trpm8^−/−^ Mice

The influence of the TRPM8 receptor on the opioid analgesic pathway has been reported, indicating that menthol has led to pain relief through the opioid pathway. Still, few studies have yet mentioned the cannabinoid pathway. This study partly clarified the relationship of the TRPM8 receptor with the cannabinoid-mediated pain relief pathway.

The Endocannabinoid System’s (ECS) involvement in several physiological regulation pathways makes it a tempting target for drugs and therapies used to treat pain and inflammation. Endocannabinoid activation reduces behavioral responses to acute, inflammatory, and neuropathic pain [60]. Given their availability, cannabinoids represent an intriguing alternative to the treatment of inflammatory pain. Although CB2-mediated effects cannot be ruled out, CB1 receptors may play a major role [61]. Preclinical studies have demonstrated the efficacy of cannabinoids in treating pain resistant to conventional treatment [61].

One of the exogenous cannabinoid compounds used is WIN 55,212-2, which provides relief from thermal and mechanical hyperalgesia in the CCI model [62]. The hot plate test revealed antinociceptive effects of WIN 55, 212-2 administered intrathecally (IT) to diabetic rats. In these inflammatory tests, the synthetic cannabinoid agonist WIN 55,212-2 (0.5, 1 mg/kg) also significantly reduced nociceptive behavioral responses in orofacial pain disorders [63].

In our study, WIN 55,212-2 alleviated inflammatory pain in both wild-type mice and *Trpm8^−/−^* mice. The noteworthy observation was that the effect of WIN 55,212-2 was stronger when the TRPM8 was suppressed in wild-type mice or the *Trpm8^−/−^* mice compared to treatment with WIN 55,212-2 alone in wild-type mice. It seems that TRPM8 activation may counteract the beneficial effects of WIN 55,212-2. This is also a premise for future efforts to improve the effectiveness of cannabinoids in clinical pain relief.

The hypothesis proposed by our study is illustrated in Figure 7. EA induces analgesia at GB34 and LR3 via the opioid and cannabinoid pathways. Activation of the opioid pathway relies on TRPM8 for significant analgesia. Combining menthol with EA boosts its analgesic efficacy. Interestingly, TRPM8 appears to inhibit the endocannabinoid pathway in wild-type mice. Inhibition or knockout of TRPM8 enhances the cannabinoid pathway, thereby augmenting analgesia.

### 3.4. Strengths and Limitations of the Proposed Hypothesis

This study hypothesizes that EA at the GB34 and LR3 acupoints induces analgesia through opioid and cannabinoid pathways, with TRPM8 playing a key modulatory role. While the findings suggest that TRPM8 activation enhances the opioid pathway in EA-induced analgesia, several strengths and limitations warrant discussion.

A key strength of this hypothesis is its potential to provide mechanistic insights into TRPM8′s role in pain modulation, particularly through its impact on opioid and cannabinoid interactions. Additionally, the enhanced analgesia observed when EA is co-applied with menthol suggests that targeted activation of TRPM8 could improve clinical EA efficacy.

However, several limitations must be addressed. First, the study uses a CFA-induced inflammatory pain model in mice, which may not fully capture the complexity of pain modulation across different types of pain. Validation in neuropathic pain or other models is needed to determine whether TRPM8′s role is pain-specific or broadly applicable. Additionally, the study focuses on EA at specific acupoints (GB34 and LR3); investigating other acupoints may strengthen the generalizability of these findings.

Further, the observed shift from opioid to cannabinoid dominance in TRPM8-deficient mice, along with partial reversal of analgesia in wild-type mice treated with AM251 and naloxone, requires clarification. A deeper understanding of each pathway’s distinct role in TRPM8′s absence would enrich the study’s mechanistic insights. Moreover, the limited discussion on translational potential should consider the differences in TRPM8 expression and function between humans and mice, as these may impact the clinical applicability of the findings. Finally, while menthol and AMTB were used as TRPM8 agonists and antagonists, dose-response studies are needed to confirm that the observed effects are due to specific TRPM8 modulation, rather than off-target effects.

Dose-response studies are crucial to confirm specificity. Future research needs to include dose-response experiments with menthol and AMTB to better characterize their effects and eliminate potential off-target interactions. By establishing dose-response curves in both wild-type and TRPM8 KO mice, we can validate that their pain-modulating effects are specifically mediated through TRPM8.

In addition, male mice were used in this study, a common choice in animal research due to several factors. Hormonal variability in female mice can introduce fluctuations affecting pain perception [64], while male mice provide greater consistency and reproducibility [65]. Historical bias [66] and behavioral differences between sexes [67] further support this choice. However, this male preference is increasingly recognized as a limitation, as sex differences are important in many physiological and pathological processes. Therefore, there is a growing movement towards including both male and female animals in studies to improve the applicability of research findings to both sexes.

In summary, while the hypothesis offers promising insights into TRPM8′s role in EA-induced analgesia, addressing these limitations in future research would enhance its clinical relevance and translational potential.

### 3.5. Clinical Relevance and Translational Potential of TRPM8 Targeting in EA Treatments

Exploring the translational potential of targeting TRPM8 in EA for human clinical use, especially in pain management, is critical. Challenges include differences in TRPM8 expression and function between humans and mice, particularly in temperature sensitivity, receptor distribution, and involvement in pain pathways. While TRPM8 mediates pain responses across species, further validation is required to clarify its role in human clinical settings. Our murine inflammatory pain model demonstrates TRPM8′s involvement in EA-induced analgesia, but human inflammatory pain conditions are more complex, with additional interactions between opioid and cannabinoid pathways. To bridge these gaps, further clinical studies regarding the effects of menthol on EA’s analgesic effect are warranted.

### 3.6. Impact of Cumulative and Long-Term Effects of EA on TRPM8 Functionality in Pain Management

Our study focuses on an acute pain model using EA to investigate the immediate effects of TRPM8 modulation on pain relief. While significant TRPM8-mediated analgesic effects were observed in the short term, repeated EA sessions could influence TRPM8 expression or activity over time. Chronic EA stimulation may alter TRPM8 expression in response to prolonged pain signaling and neuronal plasticity, which could affect the long-term efficacy of EA in managing pain.

Regarding cumulative effects, repeated EA may induce changes in immune and glial cell activity, including microglial activation. Microglia play a key role in the central nervous system’s response to injury and pain, and chronic pain can lead to persistent microglial activation, potentially influencing TRPM8 functionality. Future studies should explore how repeated EA modulates TRPM8 expression in the context of microglial activation and whether this results in enhanced or diminished analgesic effects over time.

Beyond pain, some findings suggest potential applications for TRPM8 activation in other conditions, such as ischemic stroke, where peripheral TRPM8 activation has demonstrated neuroprotective effects [68]. Future research should further elucidate the molecular mechanisms by which TRPM8 influences the balance between opioid and cannabinoid pathways and explore the clinical potential of TRPM8 modulators in diverse pain and neuroinflammatory conditions.

## 4. Material and Methods

### 4.1. Animal

Male adult C57BL/6 mice (BioLasco Taiwan Co., Ltd., Taiwan; wild type), aged between 5 and 7 weeks and weighing about 20–25 g, and *Trpm8*^−/−^ mice (strain name: B6.129P2-Trpm8^tm1Jul/J^; Jackson Laboratory, Bar Harbor, ME, USA) weighing 20–30 g and aged 6–8 weeks, were used in the experiment. The research was carried out following the regulations set by the Institutional Animal Care and Use Committees at the College of Medicine, National Taiwan University (approval number: 2021-319). The mice were kept in plastic cages, with each cage accommodating five mice, in a dedicated room with a 12-h light-dark cycle. They were provided with unlimited access to food and water throughout the study. Before commencing the experiments, the mice were transferred from their home cages to a separate behavior room and allowed to acclimate for at least one hour. All experiments were performed between 9 a.m. and 6 p.m.

### 4.2. Experimental Design

Six sets of experiments were performed in this study. In Experiment 1, the animals were randomly divided into five groups (*n* = 6): (1) naive group, (2) CFA group, (3) CFA + EA group, (4) CFA + Sham-EA group, and (5) CFA + Lido + EA group. In Experiment 2, the animals were randomly divided into four groups (*n* = 8): (1) CFA + Vehicle + EA group, (2) CFA + Naloxone + EA group, (3) CFA + AM251 + EA group, (4) CFA + Naloxone + EA + AM251 group. In Experiment 3, the animals were randomly divided into five groups (*n* = 6): (1) Vehicle + EA group, (2) AMTB + EA group, (3) AMTB + Naloxone + EA group, (4) AMTB + AM251 + EA group, and (5) AM251 + Vehicle + EA group. In Experiment 4, the *Trpm8^−/−^ mice* were randomly divided into five groups (*n* = 6): (1) CFA, (2) CFA + EA, (3) Vehicle + EA, (4) Naloxone + EA, and (5) AM251 + EA. In Experiment 5, the *Trpm8^−/−^ mice* and wild-type mice were randomly divided into two groups (*n* = 6): (1) Vehicle + EA group and (2) Menthol + EA group. In Experiment 6, the *Trpm8^−/−^ mice* and wild-type mice were randomly divided into five groups (*n* = 6): (1) Vehicle + WIN group, (2) AM251 + WIN group, (3) AMTB + WIN group, (4) *Trpm8^−/−^* Vehicle + WIN group, and (5) *Trpm8^−/−^* AM251 + WIN group.

### 4.3. Inflammation Pain Model

The mice were anesthetized using 2% isoflurane and received an injection of 30 μL of either saline (pH 7.4, buffered with 20 mM HEPES) or CFA (complete Freund’s adjuvant; 30 μg/30 μL heat-killed Mycobacterium tuberculosis, sourced from Sigma, St. Louis, MO, USA) into the plantar surface of their left hind paw to induce localized inflammation. Behavioral tests were conducted on days 1 and 2 post-inflammation induction. All tests were performed at approximately 25 °C (room temperature). Stimuli were administered only when the mice were calm and not engaged in sleep or grooming behaviors.

### 4.4. Behavioral Tests

Tests involving electronic stimulation were conducted to determine the mechanical sensitivity using a von Frey filament (EVF-3; Bioseb, Vitrolles, France) both one day before and after CFA injection. Mice were randomly placed on a metal mesh grid beneath a plastic chamber for 60 min to habituate. Following habituation, the von Frey filament was applied to each hind paw from beneath the metal mesh flooring. The von Frey test was performed in a double-blinded manner. The von Frey filament was applied perpendicularly to the plantar surface of the paw with an upward force sufficient to contact the hind paw. The von Frey filament was applied five times to each hind paw, and the mechanical withdrawal response was automatically recorded in grams. The values of the paw withdrawal thresholds from five trials were averaged [69,70].

### 4.5. Study Drugs

Complete Freund’s adjuvant (CFA, 1 mg/mL heat-killed Mycobacterium tuberculosis); AM251; Naloxone; WIN 55,212-2 (W102) were purchased from Sigma-Aldrich Co., LLC, St. Louis, MO, USA; Menthol was purchased from Alfa Aesar, Heysham, Lancashire, LA3 2XY, UK. AMTB (N-(3-Aminopropyl)-2-[(3-methylphenyl)methoxy]-N-(2-thienylmethyl)benzamide hydrochloride) was purchased from Tocris Bioscience, Bristol, BS11 9QD, UK. Lidocaine was purchased from Sigma-Aldrich (St. Louis, MO, USA). AM251, AMTB, lidocaine, and naloxone were dissolved in 0.9% saline. In addition, Menthol and WIN 55,212-2 were dissolved in DMSO. Except for Menthol, which was prepared at the working concentration for oral gavage, all other drugs were prepared at their working concentrations for i.p. injections.

### 4.6. Drugs Injection

AM251 (1 mg/kg, i.p.), Naloxone (1 mg/kg, i.p.), WIN 55,212-2 (1.5 mg/kg, i.p.), Menthol (20 mg/kg, oral gavage: 10 mL/kg), and AMTB (3 mg/kg, i.p.) were administered intraperitoneally before isoflurane anesthesia for 5 min, and lidocaine (2%, 10 μL) was administered near the GB34 acupoint (proximally 2.0 mm) [24] 1 min prior to EA treatment.

### 4.7. Electroacupuncture

The mice were given two sessions of EA on Days 1 and 2. Stainless steel acupuncture needles (32-gauge) were inserted into the defined acupoints and stimulated with electronic current provided by an Ito Trio-300 stimulator (Ito, Japan) at an intensity of 2 mA, 2 Hz, with a 150 μs pulse width in the EA group. Sham-EA group was applied to the same acupoints (GB34 and LR3) without electrical stimulation.

The murine equivalents of the human Yanglingquan (GB34) as a distal acupoint and Taichong (LR3) as a local acupoint are shown in Figure 1A. The murine Yanglingquan (GB34) is located at the fibula head that appears anteriorly and inferiorly in the depression on the lateral side of the lower leg [71]. The murine Taichong (LR3) is located on the dorsum of the foot, between the first and second metatarsal bones, in the depression distal to the junction of the two bones, underneath the Taichong pulse (the first dorsal pedal artery of the first metatarsal bone) [72].

### 4.8. Statistical Analyses

All data on nociceptive behavior assessment were presented as means ± SD (standard deviation) values for each group. One-way ANOVA, followed by Bonferroni’s post hoc test (*p* < 0.05 considered statistically significant), was used to analyze the data between groups on each day [73]. 

## 5. Conclusions

Our study demonstrates that TRPM8 is crucial for EA-induced analgesia in CFA-induced inflammatory pain, specifically through its activation of the opioid pathway. TRPM8 activation is essential for EA to achieve pain relief via opioid signaling; however, when TRPM8 is suppressed or absent, as in *Trpm8^−/−^* mice, the analgesic effect of EA shifts predominantly to the cannabinoid pathway. This dual-pathway interaction highlights mechanistic flexibility, with TRPM8-dependent opioid signaling playing a primary role in wild-type mice, while cannabinoid mechanisms compensate in the absence of TRPM8.

These insights not only deepen our understanding of EA-induced analgesia but also have significant therapeutic implications. Modulating TRPM8 could refine pain management approaches by leveraging both opioid and cannabinoid pathways more effectively.

## Figures and Tables

**Figure 1 ijms-25-13000-f001:**
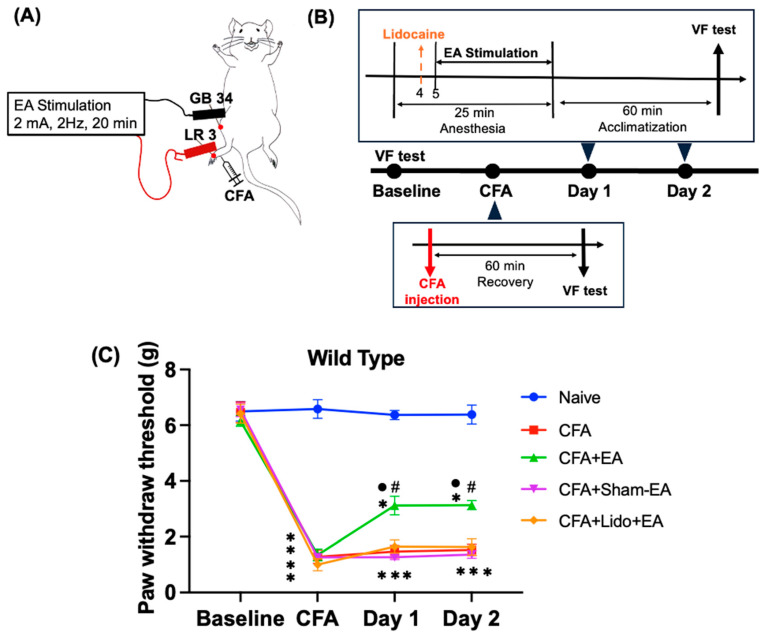
EA effectively relieves abnormal mechanical allodynia in the CFA-induced pain model. (**A**) This picture depicts the location of the acupoints used in the experiment. (**B**) Schematic flow diagram of EA in wild-type mice. (**C**) The PWT changes were measured using the von Frey test to assess mechanical allodynia in WT mice. EA was applied on Days 1 and 2, respectively. Mean and standard deviation are used to express the values (*n* = 6); * *p* < 0.05 compared with the Naive group; ^#^ *p* < 0.05, compared with the CFA group; ^●^ *p* < 0.05 compared with the CFA + Lido-EA group. Multiple post hoc comparisons with one-way ANOVA and Bonferroni’s multiple comparisons were used to analyze the data statistically.

**Figure 2 ijms-25-13000-f002:**
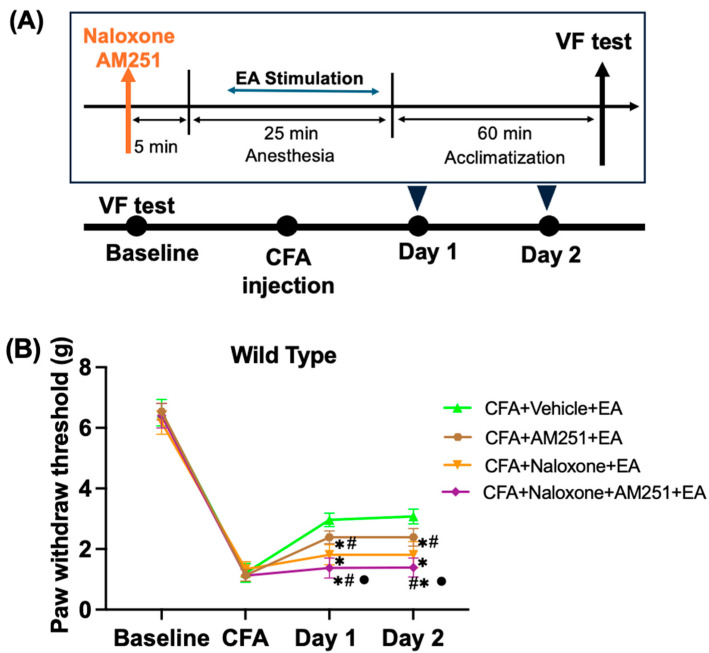
Naloxone alone or in combination with AM251 attenuates the effect of EA. (**A**) Schematic flow diagram of intraperitoneal injection (i.p.) in wild-type mice. (**B**) WT mice were tested using the von Frey test to determine changes in the PWT. On Day 1 and Day 2 after the administration of the vehicle or antagonist, EA was performed. Mean and standard deviation are used to express the values (*n* = 8). * *p* < 0.05 compared with the Vehicle + EA group; ^#^ *p* < 0.05 compared with the Naloxone + EA group; ^●^ *p* < 0.05 compared with the AM251 + EA group.

**Figure 3 ijms-25-13000-f003:**
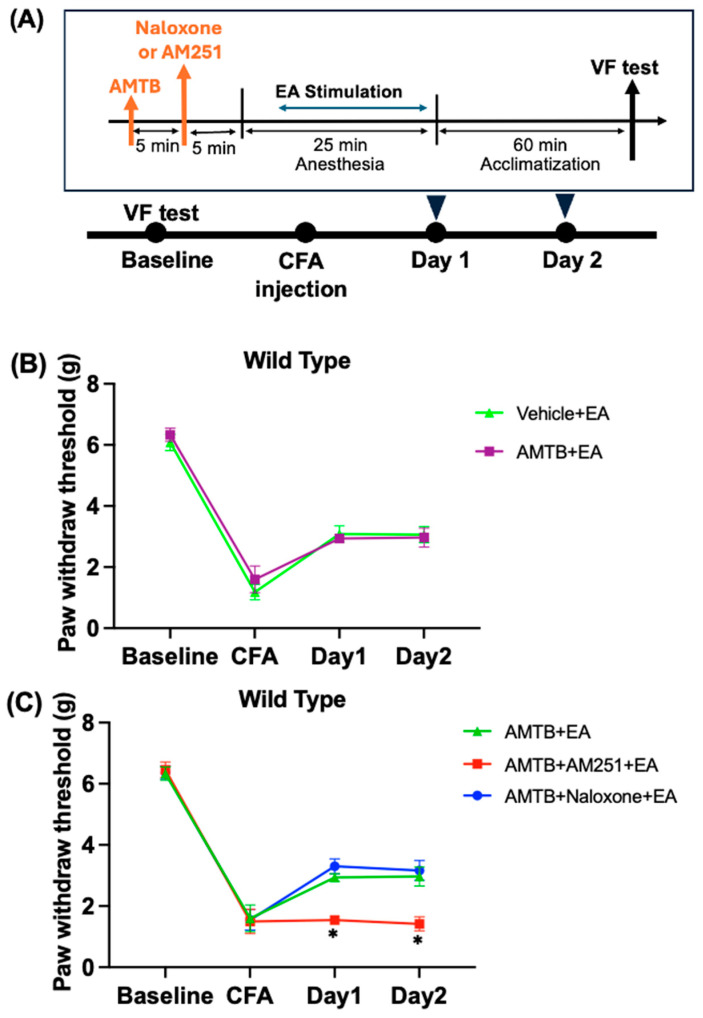
Pretreatment with AMTB had no impact on the effect of EA, which was only inhibited by AM251 and not by naloxone. The von Frey test was used to assess changes in the PWT in the Vehicle + EA group (Saline, i.p.; EA), AMTB + Naloxone + EA groups, AMTB + AM251 + EA group (AMTB + AM251, i.p.; EA), and AMTB + EA group (AMTB, i.p.; EA). (**A**) The flowchart of the experiment. (**B**) The effect of EA in the CFA pain model was assessed with AMTB (TRPM8 antagonist). (**C**) The effects of naloxone or AM251 on EA were assessed when the TRPM8 receptor was inhibited by AMTB (TRPM8 antagonist). EA was performed on Days 1 and 2 following the administration of the vehicle or antagonist. The values are presented as mean ± standard deviation (*n* = 6). * *p* < 0.05 compared with the AMTB + EA group.

**Figure 4 ijms-25-13000-f004:**
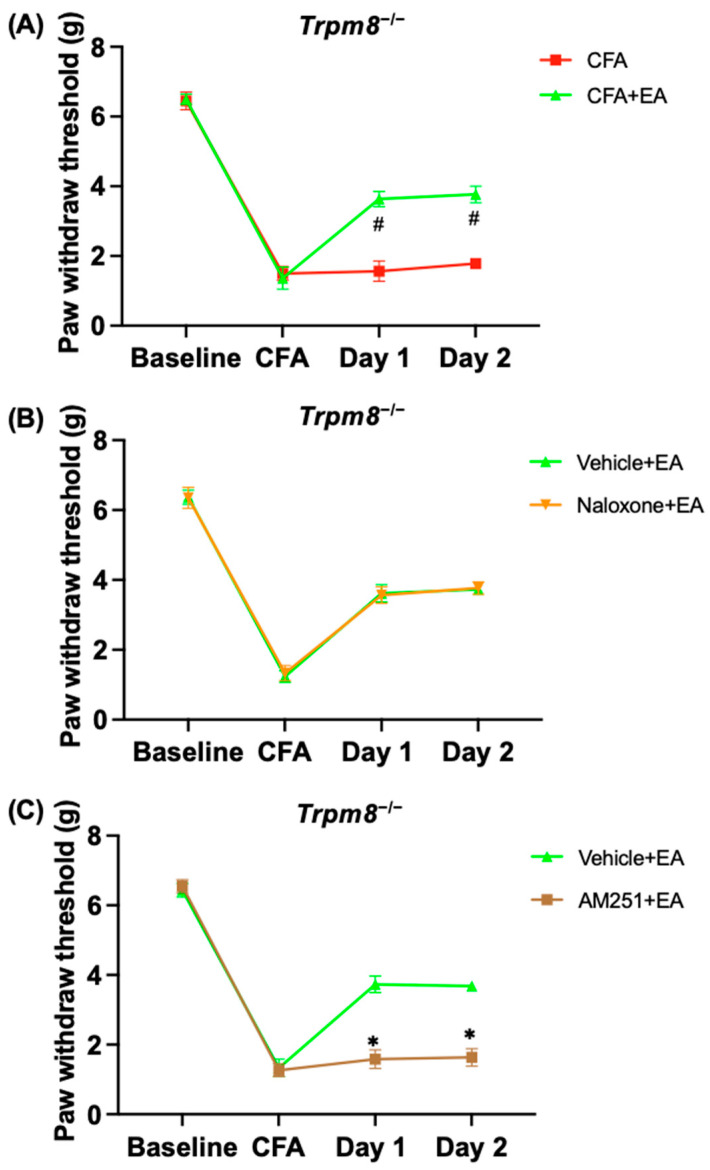
EA has an analgesic effect in the CFA pain model of *Trpm8*^−/−^ mice, which is inhibited only by AM251 but not by naloxone. In the *Trpm8*^−/−^ mice CFA pain model, the von Frey test was used to assess the PWT in the following groups: (**A**) CFA group (CFA, i.p.) and CFA + EA group (CFA, i.p.; EA). (**B**) Vehicle + EA group (Saline, i.p.; EA) and Naloxone + EA group (Naloxone, i.p. on Days 1 and Day 2 after the administration of the vehicle or antagonist; EA). (**C**) Vehicle + EA group (Saline, i.p.; EA) and AM251 + EA group (AM251, i.p. on Day 1 and 2 after the administration of the vehicle or antagonist; EA). Mean ± standard deviation (*n* = 6) is used to express the values. * *p <* 0.05 compared with the Vehicle + EA group; **^#^**
*p <* 0.05, compared with the CFA group.

**Figure 5 ijms-25-13000-f005:**
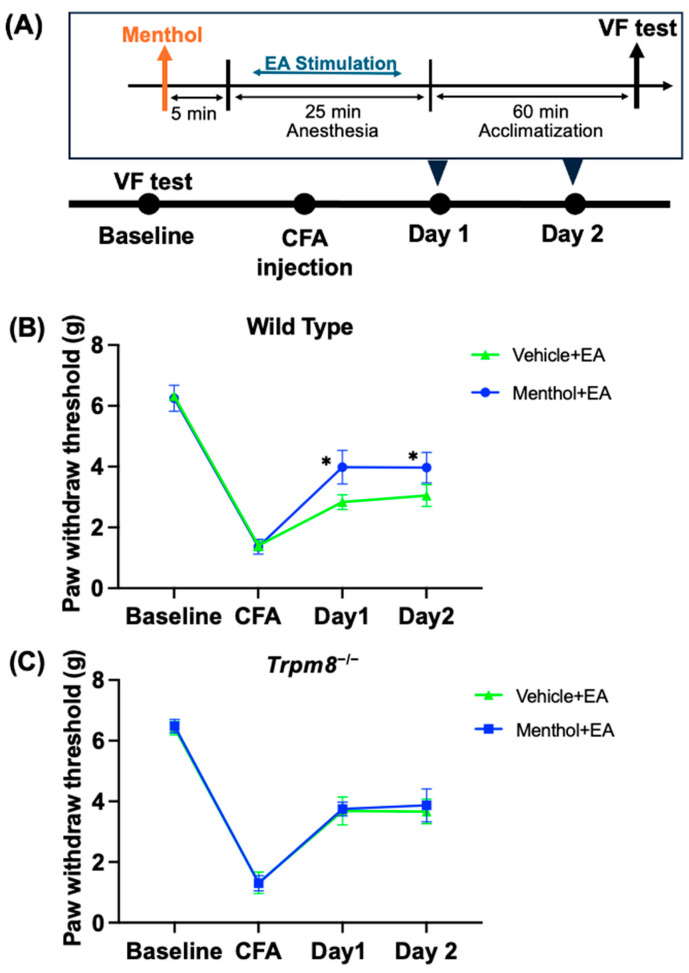
Effect of oral menthol administration on the effects of EA analgesia in wild type mice and *Trpm8*^−/−^ mice. (**A**) The flowchart depicting the experimental procedure. Using the von Frey test, the PWT of the Vehicle + EA group (Vehicle, p.o.; EA) and the Menthol + EA group (Menthol, 20 mg/kg, p.o.; EA) in the CFA pain model was assessed in (**B**) WT mice and (**C**) *Trpm8*^−/−^ mice. EA was administered on Days 1 and 2 following the administration of a vehicle or TRPM8 activator (Menthol). The values are expressed as mean ± standard deviation. * *p <* 0.05 compared with the Vehicle + EA group.

**Figure 6 ijms-25-13000-f006:**
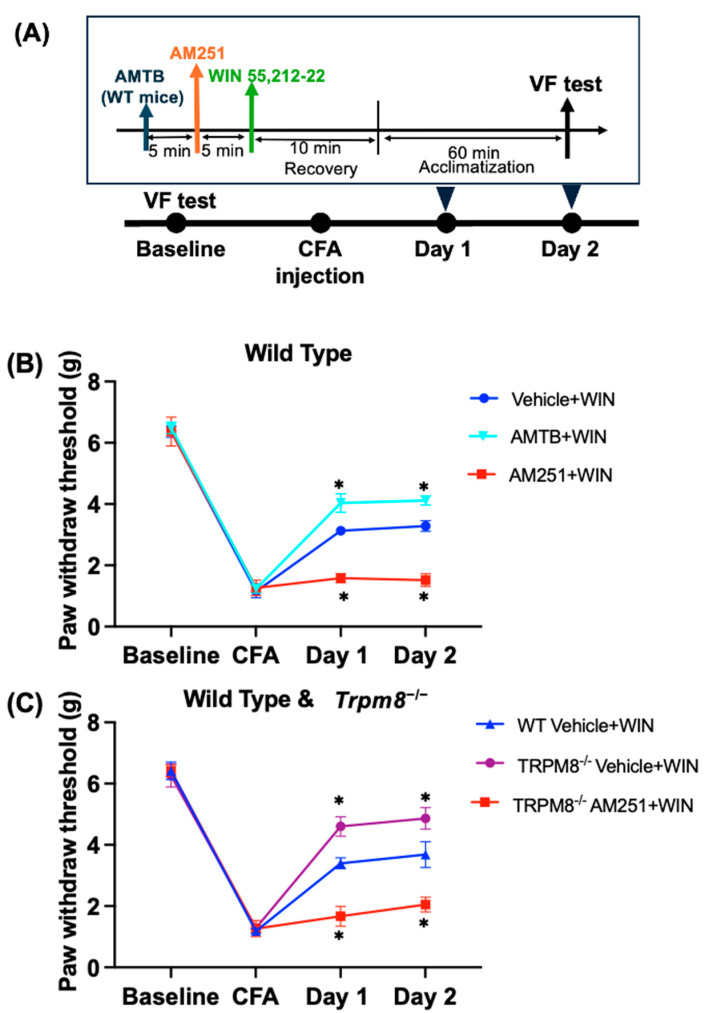
Effect of WIN 55,212-2 in the CFA pain model in wild-type and *Trpm8*^−/−^ mice. (**A**) The flowchart of the experiment. (**B**) In the CFA pain model of WT mice, the von Frey test was used to evaluate the Vehicle + WIN group (Saline + WIN 55,212-2, i.p.), AM251 + WIN group (AM251 + WIN 55,212-2, i.p.), and the AMTB + WIN group (AMTB + WIN 55,212-2, i.p.). (**C**) Changes in mechanical pain threshold of WT Vehicle + WIN group, *Trpm8*^−/−^ Vehicle + WIN group, and *Trpm8*^−/−^ AM251 + WIN group. WIN 55,212-2 treatment was performed after the administration of vehicle or antagonist on Days 1 and 2. The values are expressed as mean ± standard deviation (*n* = 6). * *p <* 0.05 compared with the WT Vehicle + WIN group.

**Figure 7 ijms-25-13000-f007:**
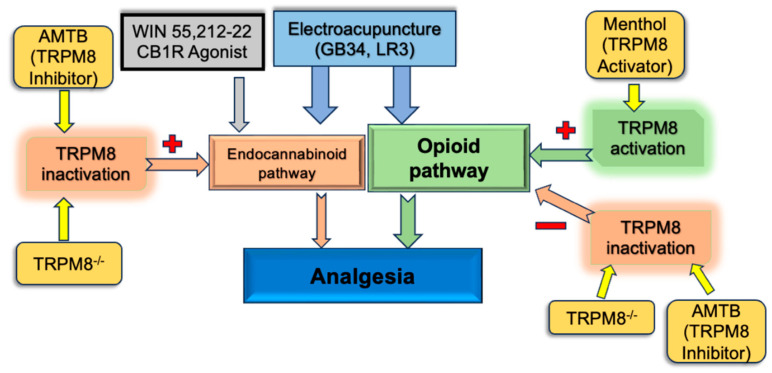
Regulatory effect of the TRPM8 receptor on acupuncture analgesia.

## Data Availability

The data that support the findings of this study are available from the corresponding author, Yi-Hung Chen, upon reasonable request.

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
