# Peer review of "TRPM8′s Role in the Shift Between Opioid and Cannabinoid Pathways in Electroacupuncture for Inflammatory Pain in Mice"

_ijms, 2024, doi:10.3390/ijms252313000_

Round 1
Reviewer 1 Report
Comments and Suggestions for Authors
The manuscript submitted by Dinh-Trong Pham et al., titled “The Role of TRPM8 in Electroacupuncture for Treating Inflammatory Pain Induced by Complete Freund's Adjuvant (CFA) in Mice,” reflects commendable scientific rigor and innovation. While the authors have laid a strong foundation, a closer examination highlights several key areas where further clarification and refinement could enhance the depth and impact of their findings. Addressing these nuanced aspects will significantly strengthen the manuscript, positioning it for well-deserved recognition and an expedited path to acceptance.
Detailed Comments
- While TRPM8 appears to facilitate opioid pathway activation in response to EA, its precise role within the nociceptive circuitry remains unclear. A more detailed discussion on how TRPM8 modulates sensory pathways during EA, potentially through neurotransmitter release or receptor sensitivity modulation, would strengthen the study.
- The study’s reliance on a murine model of CFA-induced inflammatory pain may not fully capture the complexities of pain modulation in other types of pain, such as neuropathic pain. It would be beneficial to revalidate whether TRPM8’s modulation of EA-induced analgesia is specific to inflammatory pain or applicable to broader pain conditions and rationale behind this.
- Since EA was applied specifically at GB34 and LR3 acupoints, it would be useful to clarify whether these findings are unique to these acupoints or could be generalized to others. Providing a rationale for selecting these points and discussing whether similar outcomes might be expected from other acupoints involved in pain modulation could offer valuable context.
- The shift from the opioid to cannabinoid pathway in TRPM8-deficient mice requires further clarification. Given the partial reversal of EA-induced analgesia with AM251 and naloxone in wild-type mice, a detailed breakdown of how and why each pathway contributes differently in the absence of TRPM8 would help confirm the specificity of each pathway’s involvement.
- While the study suggests that targeting TRPM8 could optimize EA outcomes, the discussion on the clinical relevance and translational potential of these findings is limited. Highlighting any challenges in translating these findings to human EA treatments, especially regarding differences in TRPM8 expression or functionality between humans and mice, would address potential concerns about real-world applicability.
- As menthol is a partial TRPM8 agonist, additional controls or dose-response experiments might be required to rule out any nonspecific effects. Similarly, for AMTB as a TRPM8 antagonist, a more detailed dose-response relationship could help confirm that observed effects are truly due to TRPM8 inhibition rather than off-target effects.
- Since the study examines how TRPM8 influences EA-induced analgesia, it would be valuable to consider how TRPM8 expression or activity may change with repeated EA sessions or under varying physiological conditions. Discussing whether EA has potential cumulative or long-term effects on TRPM8 functionality could provide additional insight into its use in pain management.
- The study exclusively used male adult C57BL/6 mice. It would be helpful to explain the rationale behind this choice and discuss whether the findings can be generalized to female mice. Addressing this could provide a broader perspective on the implications for drug development processes.
Reviewer 2 Report
Comments and Suggestions for Authors
The paper “The Role of TRPM8 in Electroacupuncture for Treating Inflammatory Pain Induced by Complete Freund's Adjuvant (CFA) in Mice“ is well-written presentation of an experimental research conducted on animal model. It should be considered for publication in IJMS after few minor modifications.
1. This paper presents an innovative research about electroactupuncture in the treatment of inflammatory pain in animal model with new hypothesis which cannot be seen in the title. So would suggest some changes in the title in order to emphasize the new hypothesis about the TRPM8 and the assumed mechanism of action of electroacupuncture in inflammatory pain treatment.
2. The introduction is informative and gives an insight in all the most important terms that are important for further understanding of the experiment. Clear aim should be defined.
3. Compliments for the schematic presentation and the figures given in the paper which help in understanding the experiment.
4. Limitations and strength of the proposed hypothesis given in Figure 7 should be given at the end. Otherwise the disscussion part is well-written, clear and argumentative.
Reviewer 3 Report
Comments and Suggestions for Authors
Please see the attached file.

The English could be improved to more clearly express the research.
Round 2
Reviewer 3 Report
Comments and Suggestions for Authors
The authors have responded to my remarks. I have no further questions.
Comments on the Quality of English LanguageIt is good.